# Frequency of ventricular arrhythmias in apparently healthy, large breed dogs during seven-day Holter monitoring

Tamilselvam Gunasekaran  *, Alyssa Pinkos, Robert Sanders

Department of Small Animal Clinical Sciences, College of Veterinary Medicine, Michigan State University, East Lansing, Michigan, United States of America

* tamilsel@msu.edu

## Abstract

Continuous ambulatory electrocardiographic monitoring (Holter) is commonly used to diagnose and manage various cardiac arrhythmias in dogs. Despite its widespread use, data on the frequency of ventricular arrhythmias and its day-to-day variability in healthy, large-breed dogs without known predisposition to cardiomyopathy remains limited. This study assessed the frequency, complexity, and spontaneous day-to-day variation of ventricular arrhythmias in clinically healthy, large-breed dogs using seven-day Holter monitoring. Thirty-one, apparently healthy dogs without any history of systemic illness and normal physical examination findings underwent continuous 7-day Holter monitoring. Two dogs were excluded due to the occurrence of significant systemic disease soon after enrolment. The results from 29 dogs showed that most dogs (86%) had fewer than twenty ventricular premature complexes (VPCs) per 24-hour period, with significant day-to-day variation (up to 93%) in 4 dogs with over twenty VPCs. Complex ventricular arrhythmias, such as ventricular couplets, triplets, and ventricular tachycardia, were rare. No correlation was found between age and VPC frequency (p = 0.409) in this population of predominantly older dogs. These findings suggest that large-breed dogs without a predisposition to cardiomyopathies exhibit low arrhythmia frequencies with significant day-to-day variation in some dogs.

## Introduction

Continuous ambulatory electrocardiographic recordings (Holter) are used for the diagnosis and management of various cardiac arrhythmias in dogs [1,2]. Holter recordings can provide prognostic information [3,4], predict the risk of future disease development [3], and help guide anti-arrhythmic drug therapy [5,6]. Additionally, Holter monitoring is one component of the gold standard screening to identify occult cardiomyopathies in clinically healthy Boxers and Doberman Pinschers [7–9]. Despite its widespread use in veterinary cardiology practices, information about the prevalence, frequency, and complexity of arrhythmias in normal dogs is mostly available

**Data availability statement:** All relevant data are within the manuscript and its Supporting Information files.

**Funding:** The author(s) received no specific funding for this work.

**Competing interests:** The authors have declared that no competing interests exist.

for breeds predisposed to developing cardiomyopathies [10–13]. Generalization of the findings from breed-specific studies to all dog breeds can be problematic. In humans, ventricular arrhythmias can be seen in healthy adults and the frequency of such arrhythmias increases with age [14–16]. Additionally, day-to-day spontaneous variation in ventricular arrhythmias has been reported in human patients [17,18]. A large study analyzed 14-day Holter recordings from over 8,000 human adults and found substantial day-to-day fluctuations in the frequency of ventricular premature complexes (VPCs) [19]. Specifically, after 7 days of monitoring, the estimated daily VPC frequency differed by 50% or more from the 14-day average in 25% of participants [19]. The spontaneous day-to-day variation in the frequency and complexity of ventricular arrhythmias has also been reported in Boxers and Doberman Pinschers [9,12]. In Boxers, the spontaneous day-to-day variability of VPC frequency was 80% in those dogs that had more than 500 VPC's in 24 hours [12]. One screening study performed in healthy Doberman Pinschers concluded that 4 days of Holter monitoring was required to identify the predictive criteria for dilated cardiomyopathy due to significant day-to-day variation in VPC frequency [9]. A few veterinary studies have assessed the frequency of ventricular arrhythmias in healthy adult dogs and puppies [10,20–23]. However, existing studies rely on 24-hour Holter recordings, and the spontaneous day-to-day variation in ventricular arrhythmias has not been characterized through a prospective study in healthy, large-breed dogs without a predisposition to arrhythmias. The objectives of this study are to report the frequency and complexity of ventricular arrhythmias and their day-to-day spontaneous variation in clinically healthy, large-breed dogs using 7-day Holter monitoring.

## Materials and methods

Clinically healthy dogs over 22 Kg owned by students and employees at the Veterinary Medical Center at Michigan State University were prospectively assessed for eligibility in the study. Each dog underwent a comprehensive history review and physical examination conducted by a board-certified cardiologist as part of the screening process. Dogs with normal physical examination results were included in the study. Breeds known to have a predisposition to ventricular arrhythmias, including Boxers, Doberman Pinschers, Great Danes, and German Shepherds, were excluded from the study. Additionally, dogs with known heart disease, abnormal cardiac auscultatory findings, or those receiving medications that could affect heart rate and or rhythm were excluded from the study. Informed consent was obtained from the owners, and the study protocol received approval from the Institutional Animal Care & Use Committee at Michigan State University.

A 2-channel, transthoracic Holter monitor (LifeCard CF, Spacelabs Healthcare, Snoqualmie, Wisconsin, United States of America) capable of recording 7 continuous days was applied on each dog and secured using a fitting vest [2]. No other changes from the dog's normal routine in diet, exercise, or environmental stimuli were requested. The dogs were discharged with their owners to their home environment. The owners were trained on maintaining the recording system and were informed to change all electrode patches once after 3–4 days. The Holter monitors were removed

by the owners after 7 continuous days of recording. The returned Holter recordings were analyzed using automated computer software (Pathfinder SL, Spacelabs Healthcare, Snoqualmie, Wisconsin, United States of America) with analyses verified by a board-certified veterinary cardiologist. This verification process involved the detection of arrhythmias that were identified as normal complexes by the automated system and vice versa, assessment for normal triggering, removal of artifacts, and manual verification of arrhythmic events. Dogs that had more than five percent artifacts on their Holter recording were removed from the study. A preset criteria were used for the classification of arrhythmias by the Holter analysis software (see Table 1).

For each 24 hours of the recording, the mean, minimum, and maximum heart rate (HR) were determined. The frequency of ventricular arrhythmias was tabulated. Additionally, the complexity of the ventricular arrhythmia was noted by recording the number of couplets, triplets, ventricular tachycardias, and the occurrence of R on T beats observed. The total number of pauses per day was recorded along with the maximum duration. Statistical analysis was performed using commercial statistical software (R studio, The R Foundation for Statistical Computing, Vienna, Austria). Statistical significance was set at an alpha value of 0.05. Normality was tested using the Anderson-Darling test. Normally distributed variables were reported as means and standard deviations. Non-normally distributed variables were reported as medians and ranges. The relationship between the number of VPCs and age was tested using Spearman's Rank correlation coefficient.

## Ethics statement

This study was conducted in compliance with ethical standards for animal research. Approval was obtained from the Institutional Animal Care & Use Committee (IACUC) at Michigan State University before study initiation. Informed consent was obtained from all dog owners before participation.

## Results

A total of 31 dogs participated in the study and underwent 7-day Holter monitoring. Two of the dogs were excluded from the final analysis (see below). The final cohort included 21 mixed-breed dogs, 2 Labrador Retrievers, 2 Border Collies, 1 Dogue de Bordeaux, 1 Alaskan Malamute, 1 Basset Hound, and 1 Weimaraner. The median age and body weight were 10 years (range: 2–12 years) and 28.4 Kg (range = 22.1 to 42.5 Kg), respectively. Six dogs were younger than 5 years, five dogs were between 6 and 8 years, and eighteen dogs were older than 8 years. All dogs had at least 5 days of Holter recording available for analysis. The median recording duration was 6 days and 21 hours (range = 5 days and 3 hours to 7 days and 1 hour).

One mixed-breed dog, which underwent a normal physical examination during initial study enrolment, was diagnosed with hemoperitoneum due to a splenic mass within 3 weeks of Holter monitoring. This dog exhibited 1909 isolated VPCs

**Table 1. Criteria used for seven-day Holter analysis in apparently healthy large breed dogs.**

| Arrhythmia/beat classification | Classifier used |
|---|---|
| Ventricular premature complexes | < 100% of prevailing RR[a] interval |
| Ventricular escape beats | >= 150% of prevailing RR[a] interval |
| R on T[b] | < 170ms + 5% of prevailing RR[a] interval |
| Idioventricular rhythm | >= 4 beats <= 50 bpm |
| Accelerated idioventricular rhythm | >= 4 beats between 51 and 180 bpm |
| Ventricular tachycardia | >= 4 beats >= 181 bpm |
| Ventricular bigeminy and trigeminy | >= 4 cycles |
| Pause | >= 2.00 seconds |

[a]RR interval: interval between two consecutive R waves during normal sinus rhythm.

[b]R on T: R waves occurring on top the T wave of the preceding QRS complex.

during a 7-day Holter recording, including one episode of ventricular tachycardia (VT) at a heart rate (HR) of 232 beats per minute (bpm). An additional dog (Golden Retriever) had 1889 isolated VPCs during his 7-day Holter recording, including one episode of VT at an HR of 188 bpm. This dog was euthanized within a month of the Holter evaluation due to an unknown neurologic disease. Consequently, both these dogs were excluded from further analysis due to the potential contributions of systemic disease to ventricular arrhythmias. Therefore, Holter recordings from 29 dogs were available for final analysis. The summary of Holter findings from the seven-day Holter recording is depicted in Table 2.

When considering the maximum number of VPCs per 24 hours of the Holter recording, most dogs had an infrequent number of VPCs (see Table 3). Due to the low frequency of VPCs in study dogs, the spontaneous day-to-day variation in VPC frequency was only calculated for 4 dogs that had more than 20 VPCs in 24 hours. In these 4 dogs, the spontaneous day-to-day variation

$$(Maximum\ daily\ VPC\ frequency - minimum\ daily\ VPC\ frequency)/maximum\ daily\ VPC\ frequency$$ranged from 50% to 93%. Fourteen dogs (56% or 17 out of 30 dogs) had at least one couplet (median = 1 couplet, range = 0–12 couplets) in a 24-hour recording period. No episodes of ventricular bigeminy or trigeminy were noted.

Five dogs (16% (3 out of 30 dogs)) had one triplet (median = 0 triplets, range = 0–1 couplet) each in a 24-hour recording period. In 6 out of 17 dogs that had the couplets, no couplets were noted on day 1 of the Holter recording. Similarly, in 1 out of 3 dogs where triplets were noted, no triplets were noted on day 1 of the Holter recording. One dog had one episode of VT (223 bpm) and one episode of accelerated idioventricular rhythm (144 bpm), and one dog had 2 episodes of VT (224 bpm and 262 bpm) during a 24-hour recording period. None of these episodes were noted on the 1st of the Holter recording. There was no significant correlation between the number of VPCs to the dog's age (p-value = 0.409;

**Table 2. Parameters from the entire seven-day Holter recording in 29 healthy large-breed dogs.**

| Holter Parameters | Mean ± standard deviation or median (range) (Seven-days) |
|---|---|
| Total number of ventricular complexes | 62 ± 112 |
| Number of premature ventricular complexes | 40 ± 68 |
| Number of ventricular escape complexes | 1 (0–399) |
| Number of couplets | 1 (0–26) |
| Number of triplets | 0 (0–2) |
| Number of ventricular tachycardia | 0 (0–2) |
| Number of accelerated idioventricular rhythm | 0 (0–1) |
| Mean 24-hour heart rate (bpm)[c] | 72 ± 8 |
| Maximum heart rate[d] (bpm)[c] | 204 ± 25 |
| Minimum heart rate[d] (bpm)[c] | 40 ± 6 |
| Longest pause duration (in seconds) | 3.8 ± 0.8 |

[c]beats per minute.

[d]The maximum and minimum HR is based on the maximum one-minute HR for each dog during the 7-day Holter recording.

**Table 3. Maximum number of ventricular premature complexes per 24-hour period of the 7-day Holter recording in clinically healthy adult breed dogs.**

| Maximum number of ventricular premature complexes per 24 hours | Number of dogs |
|---|---|
| 0–10 | 23 |
| 10 - 20 | 2 |
| 20 - 50 | 2 |
| 50 - 100 | 2 |

Spearman's R = 0.154). Detailed information on various ventricular complexes and their complexities per 24-hour period for each dog and their respective frequencies are available in supporting information in the S1 Table.

## Discussion

The results of this study indicate that clinically healthy, large-breed dogs exhibit a low frequency of ventricular arrhythmias, with most dogs (86%) having less than 20 VPCs in a 24-hour recording period. This finding is similar to previous studies that reported VPC frequency using 24-hour Holter monitoring in adult dogs and puppies [10,22,23]. In contrast, previous studies in healthy dogs from breeds predisposed to cardiomyopathies have consistently demonstrated a higher frequency of ventricular arrhythmias compared to our study [7,11]. Additionally, in 4 dogs that had more than 20 VPCs per 24-hour recording period, a spontaneous day-to-day variation in the frequency of VPCs as high as 93% was noted in this study. The spontaneous day-to-day variation in ventricular arrhythmias has been previously reported in clinically healthy Doberman Pinschers and Boxers [9,12]. In these breeds predisposed to cardiomyopathies, the presence of spontaneous day-to-day variation in VPC frequency significantly affects the diagnostic accuracy of 24-hour Holter monitoring in classifying risk for dilated cardiomyopathy [9,12]. However, a recent study reported that the use of 48-hour Holter recording only marginally improved the detection of clinically relevant ventricular arrhythmias in all dogs as compared to 24-hour Holter recording [5]. As such, the clinical relevance of this finding remains unknown in otherwise healthy large-breed dogs, given the lack of long-term follow-up studies assessing the relationship between spontaneous day-to-day variation in VPC frequency and disease development or prognosis. Similar to previous studies of otherwise normal dogs, dogs in this study had a rare number of ventricular arrhythmia complexities, such as couplets, triplets, and VT episodes [10,22]. In this study, a high degree of day-to-day variation in the occurrence of complex ventricular arrhythmias was also noted. Some dogs in this study had up to 399 ventricular escape complexes over the seven days of the Holter recording period. In previous studies of Holter evaluations in healthy dogs, the number of ventricular escape complexes has not been characterized except for a study performed in healthy Saluki dogs [24]. This study reported a similar number of ventricular escape complexes to our study, with some dogs having as high as a median of 612 ventricular escape complexes in 24 hours [24]. A previous study in puppies and younger adult dogs reported lower incidences of arrhythmias in puppies compared to older dogs [10,22]. Additionally, a significant effect of age was noted on Holter-obtained HR in a different study [25]. In our study, we found no significant correlation between age and VPC frequency in this study (p = 0.409). However, this discrepancy may also be due to the small sample size of younger dogs in our study.

Our study has several limitations. The sample size was relatively small (n = 29 after exclusions), including only a small number of younger dogs, which may limit the generalizability of the findings to young dogs. Additionally, despite efforts to exclude dogs with known systemic illness or auscultatory abnormalities, subclinical cardiac or systemic illness may have been present in some dogs. This is highlighted by the fact that one dog developed hemoperitoneum due to a splenic mass, and another was euthanized due to neurological disease soon after study enrolment. Although these dogs were excluded from the final analysis, their high VPC counts highlight the potential impact of systemic diseases on arrhythmia prevalence. Additionally, other dogs who had only a few complex ventricular arrhythmias, couplets/triplets, or ventricular tachycardia episodes in this study could also have subclinical cardiac or systemic disease that was not obvious on normal physical examination. Therefore, the lack of a thorough diagnostic workup can be a limitation of this study. Future studies should also consider incorporating additional diagnostic evaluations to rule out occult cardiac and or systemic illnesses. Finally, the study population primarily consisted of mixed-breed dogs, with only a few purebred dogs included. This could limit the ability to generalize findings to specific large-breed populations that may have unique cardiac characteristics.

## Conclusions

This study of predominantly older, clinically healthy large-breed dogs found a low frequency of ventricular arrhythmias, with spontaneous day-to-day variation observed in some dogs. These findings provide baseline data for interpreting Holter

recordings in large-breed dogs that are not predisposed to cardiomyopathies. Additionally, longitudinal studies assessing long-term arrhythmic trends in healthy dogs could help refine screening and diagnostic approaches in large-breed dogs, not known to be predisposed to cardiomyopathies.

## Supporting information

**S1 Table. Individual 24-hour ventricular arrhythmia frequency for each dog.**
(PDF)

## Author contributions

**Data curation:** Tamilselvam Gunasekaran.

**Formal analysis:** Tamilselvam Gunasekaran.

**Investigation:** Tamilselvam Gunasekaran, Robert Sanders, Alyssa Pinkos.

**Methodology:** Robert Sanders, Alyssa Pinkos.

**Project administration:** Robert Sanders, Alyssa Pinkos.

**Supervision:** Robert Sanders.

**Writing – original draft:** Tamilselvam Gunasekaran.

**Writing – review & editing:** Robert Sanders, Alyssa Pinkos.

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
