## [Decision Letter · Decision Letter 0]

13 Apr 2025

PONE-D-25-07264Frequency of ventricular arrhythmias in apparently healthy, large breed dogs during seven-day Holter monitoring.PLOS ONE

Dear Dr. Gunasekaran,

Thank you for submitting your manuscript to PLOS ONE. After careful consideration, we feel that it has merit but does not fully meet PLOS ONE’s publication criteria as it currently stands. Therefore, we invite you to submit a revised version of the manuscript that addresses the points raised during the review process.

We look forward to receiving your revised manuscript.

Kind regards,

Martin E. Matsumura, MD

Academic Editor

PLOS ONE

Journal Requirements:

Additional Editor Comments:

Please address all comments included in the reviewer's attached comments, thank you.

Reviewers' comments:

Reviewer's Responses to Questions

**Comments to the Author**

1. Is the manuscript technically sound, and do the data support the conclusions?

Reviewer #1: Partly

2. Has the statistical analysis been performed appropriately and rigorously? 

Reviewer #1: I Don't Know

3. Have the authors made all data underlying the findings in their manuscript fully available?

Reviewer #1: Yes

4. Is the manuscript presented in an intelligible fashion and written in standard English?

Reviewer #1: Yes

5. Review Comments to the Author

Reviewer #1: A nice study and well written. Please see attachment for comments. Some of the stats were done on small numbers of dogs and therefore we need to be careful about drawing conclusions and acknowledge the limitations. In areas where the results different from previous work, results may need to be discussed more fully in the discussion section.

6. PLOS authors have the option to publish the peer review history of their article (what does this mean? ). If published, this will include your full peer review and any attached files.

**Do you want your identity to be public for this peer review?** For information about this choice, including consent withdrawal, please see our Privacy Policy .

Reviewer #1: No

---

## [Author Response · Author response to Decision Letter 1]

18 Apr 2025

Dear reviewers and editors,

Thank you for sharing your valuable feedback and suggestions to improve our manuscript. We have clarified and made changes to the manuscript as needed. Hopefully, this will satisfy your concerns.

Frequency of ventricular arrhythmias in apparently healthy, large-breed dogs during 7-day Holter monitoring

Introduction

1. Line 24 – The variability was taken from just 4 dogs. I understand the point you’re trying to make here but we also have to be careful not to over-emphasise a single result as this could mislead the casual reader. I would be happier if you changed it to: significant day-to-day variation (50-93%) in 4 dogs with over twenty…

We understand the concern and have made the requested change.

2. Line 26-27 – I think your statement about age and frequency of vent arrhythmias is quite bold, given that the study population was predominantly older dogs. Perhaps soften this to: No correlation was found between age and VPC frequency (P = 0.409) in this population of predominantly older dogs.

We understand your concern and have made the requested change.

3. Line 40 – ref 4 – would be ok if placed next to ref 1 but this study did not investigate anti-arrhythmic drug therapy

Our apologies for the oversight, and we have made the requested change.

4. Line 41 – gold standard screening would also involve echocardiography so consider changing to: ambulatory ECG is one component of gold standard screening to identify…

• Line 46 – consider adding refs for Dobermanns, Great Danes. Maybe these ones: Currently ref 18 - Gunasekaran, T et al. “Comparison of single- versus seven-day Holter analysis for the identification of dilated cardiomyopathy predictive criteria in apparently healthy Doberman Pinscher dogs.” Journal of veterinary cardiology : the official journal of the European Society of Veterinary Cardiology vol. 27 (2020): 78-87. doi:10.1016/j.jvc.2020.01.003

• El Sharkawy, S et al. “Long-term outcome and troponin I concentrations in Great Danes screened for dilated cardiomyopathy: an observational retrospective epidemiological study.” Journal of veterinary cardiology : the official journal of the European Society of Veterinary Cardiology vol. 47 (2023): 1-13. doi:10.1016/j.jvc.2023.03.003

Recommended references are added, and the wording is changed as requested

5. Line 50 – ref 14 – not in English. Perhaps a bit harsh and you may be fluent in German but if you haven’t read the full text can you justify use as a reference?

We see your concern. Google Translate helped us understand the results and use as a reference. For the interest of other readers, we have removed that reference now.

6. Line 67 – ref 4 does look at a range of dog breeds, although I appreciate your study is prospective.

We made some changes to the wording to correctly reflect this.

7. Line 103 / table 1 - Why did you classify idioventricular rhythm as >4 beats at <50bpm and, more controversially, VT as >4 beats at >50bpm. If you stick with idioventricular rhythm at <50bpm, do you then add accelerated idioventricular rhythm for 50- 180bpm and VT for >180bpm?

We understand your reasoning for classifying ventricular ectopy and agree with this assessment. We have now included the classification of AIVR. There was only one episode of AIVR in this study. To avoid any confusion, we made sure to give the actual rates of VT witnessed in the study so that the reader is clear on the frequency and rates of VT seen in this study.

8. Lines 110-113 – how did you classify trigeminy/bigeminy?

Both the trigeminy and bigeminy were more than 4 cycles, but we did not see any in the study dogs. We have added this to the results section.

9. Line 137 – The predominance of older dogs in this study is a possible limitation

We have updated this in the limitations.

10. Lines 138-140 – the total number of dogs here is 29, not 31. I know 2 dogs were censored, but this is not clear until lines 142-152. Consider rewriting so the numbers work.

We have rewritten this section to make the numbers flow better

11. Line 155 – table title – considering stating that these data are from 29 apparently healthy large breeds…

We have changed the title.

12. Line 156 – Table 2 – I would consider moving the (seven days) to above the right hand collum containing the numbers.

Recommended change made

13. Would you consider the dog with 399 ventricular escape beats to be normal? On reflection, could this dog have sinus node dysfunction?

We agree with this statement and that a degree of sinus node dysfunction cannot be ruled out. However, a similar number of pauses were also seen in healthy saluki dogs.

Sanders RA, Kurosawa TA, Sist MD. Ambulatory electrocardiographic evaluation of the occurrence of arrhythmias in healthy Salukis. J Am Vet Med Assoc. 2018 Apr 15;252(8):966-969. doi: 10.2460/javma.252.8.966. PMID: 29595391.

Historically number of escape complexes was not well reported in normal dogs. Also, during analysis, we have ensured that these escapes were not inappropriate for the surrounding rhythm (i.e., they did not occur or were characterized by preceding tachycardias). We have included this in the discussion as well.

14. Line 167 – I’m not sure how meaningful this is when taken from only 4 dogs.

Agree that this finding in 4 dogs is not a major finding. We have updated any overstatements of this result.

15. Lines 173-181 – were these dogs with complex ectopy normal?

Finding complex ectopy is probably one of the interesting findings of this study. While rare, the occurrence of any complex ventricular arrhythmia probably warrants further investigation of potential causes. However, given the inclusion criteria of normal dogs based on normal cardiac physical examination, including lack of heart murmur, we would have difficulty retrospectively classifying these dogs as abnormal vs normal variants. We have added additional clarifications and emphasis on why this is a limitation in this study including the need for additional work up.

16. Line 179 – see the previous comment about the rate of VT – for me, 144 bpm would be an accelerated idioventricular rhythm.

We have addressed this in the previous comment.

17. Line 181 and also 213 – the way the text reads to me is quite dismissive about the effects of age but, in my opinion, this relatively small study with predominantly older dogs is not best placed to fully evaluate the effect of age on the freq of ventricular arrhythmias so perhaps consider phrasing a bit more cautiously. Consider adding this ref: Cruz Aleixo, Amanda Sarita et al. “Scaling Relationships Among Heart Rate, Electrocardiography Parameters, and Body Weight.” Topics in companion animal medicinevol. 32,2 (2017): 66-71. doi:10.1053/j.tcam.2017.06.002

Reference added and wording is altered to reflect the small sample size of young dogs.

18. Lines 194 - 196 – Compare this with the results described in Ref 4

This information is added now

19. Line 219 – I would also consider the dogs with complex ventricular ectopy to be abnormal.

See the response to the comment above

---

## [Editor Report · Decision Letter 1]

2 May 2025

Frequency of ventricular arrhythmias in apparently healthy, large breed dogs during seven-day Holter monitoring.

PONE-D-25-07264R1

Dear Dr. Gunasekaran,

We’re pleased to inform you that your manuscript has been judged scientifically suitable for publication and will be formally accepted for publication once it meets all outstanding technical requirements.

Kind regards,

Martin E. Matsumura, MD

Academic Editor

PLOS ONE

Additional Editor Comments (optional):

All reviewer comments have been addressed, thank you for the thorough and organized author response

---

## [Editor Report · Acceptance letter]

PONE-D-25-07264R1

PLOS ONE

Dear Dr. Gunasekaran,

I'm pleased to inform you that your manuscript has been deemed suitable for publication in PLOS ONE. Congratulations! Your manuscript is now being handed over to our production team.

Kind regards,

on behalf of

Dr. Martin E. Matsumura

Academic Editor

PLOS ONE